# Characterization of Rhesus Macaque Embryonic Stem Cells in Primed and Naïve-like Cell States of Pluripotency Using Fourier Transform Infrared (FTIR) Microspectroscopy

**DOI:** 10.3390/ijms26199514

**Published:** 2025-09-29

**Authors:** Jittanun Srisutush, Worawalan Samruan, Preeyanan Anwised, Anaïs Amzal, Cloé Rognard, Pierre Savatier, Irene Aksoy, Kanjana Thumanu, Rangsun Parnpai

**Affiliations:** 1Embryo Technology and Stem Cell Research Center, School of Biotechnology, Institute of Agricultural Technology, Suranaree University of Technology, Nakhon Ratchasima 30000, Thailand; jittanunsrisutush@gmail.com (J.S.); wsamruan@gmail.com (W.S.); preeyanan.biochem@gmail.com (P.A.); 2Univ Lyon, Université Lyon 1, INSERM, Stem Cell and Brain Research Institute U1208, 69500 Lyon, France; anais.amzal@inserm.fr (A.A.); cloe.rognard@inserm.fr (C.R.); pierre.savatier@inserm.fr (P.S.); irene.aksoy@inserm.fr (I.A.); 3PrimaStem Platform, Univ Lyon, Université Lyon 1, INSERM, Stem Cell and Brain Research Institute U1208, 69500 Lyon, France; 4Synchrotron Light Research Institute (Public Organization), Muang, Nakhon Ratchasima 30000, Thailand

**Keywords:** Fourier transform infrared, rhesus macaque, embryonic stem cells, primed states, naïve-like cell states

## Abstract

We evaluated the potential of Fourier-transform infrared (FTIR) microspectroscopy for non-invasive biochemical profiling of rhesus macaque embryonic stem cells (rhESCs) cultured in either conventional FGF2/KOSR medium or a novel formulation, ALGöX. Cells from both conditions were analyzed by immunocytochemistry, RNA sequencing, and high-resolution FTIR profiling. Molecular marker expression patterns and transcriptional profiles revealed that rhESCs maintained in FGF2/KOSR were in the primed pluripotent state, whereas those cultured in ALGöX adopted a naïve-like state. FTIR spectra showed consistent differences in protein, lipid, and nucleic acid signatures, with ALGöX-cultured cells displaying higher amide I/II and nucleic acid absorbance and FGF2/KOSR-cultured cells exhibiting stronger lipid-associated bands. Principal component analysis (PCA) separated the two groups along PC−1 (64% variance), and partial least squares discriminant analysis (PLS-DA) classified samples with 100% specificity and 100% sensitivity. These findings demonstrate that FTIR microspectroscopy can reliably discriminate pluripotent state–specific biochemical features in non-human primate PSCs, providing a rapid and label-free approach for monitoring stem cell identity and quality.

## 1. Introduction

Pluripotent stem cells (PSCs), derived from the epiblast of mammalian blastocysts, have the dual capacity for unlimited self-renewal and differentiation into derivatives of all three germ layers [1]. In rodents, PSCs can exist in two distinct states: the naïve state, represented by embryonic stem cells (ESCs) derived from the preimplantation epiblast, and the primed state, represented by epiblast stem cells (EpiSCs) derived from post-implantation epiblast tissue [2]. These states are also recognized in human and non-human primates (NHPs), including Rhesus macaques (*Macaca mulatta*), where PSCs can be stabilized in either a naïve and primed configuration depending on culture conditions and signaling inputs [3]. Naïve and primed PSCs differ markedly in their transcriptional, epigenetic, and metabolic profiles, which in turn have a profound impact on their functional properties [4]. Notably, only naïve PSCs in the naïve state exhibit the capacity to integrate into host preimplantation embryos and contribute to development, as demonstrated in rodents [2], rabbits [5], and cynomolgus monkeys [6]. These functional differences are underpinned by distinct regulatory networks and chromatin landscapes: naïve PSCs display global DNA hypomethylation, two active X chromosomes in female cells, and reduced levels of H3K27me3 at developmental gene loci, whereas primed PSCs show higher DNA methylation, one inactive X chromosome (in females), and accumulation of repressive histone marks [4]. Identifying the pluripotency state of a given PSC line traditionally requires multimodal analyses, including bulk RNA sequencing [7], immunostaining [8] for state-specific markers (e.g., KLF4, TFCP2L1 for naïve; OTX2, ZIC2 for primed), reporter-based functional assays, and epigenomic profiling. These approaches, while informative, are often labor-intensive, and not readily applicable for routine quality control. In this context, Fourier-transform infrared (FTIR) spectroscopy offers an attractive, non-invasive alternative for cell characterization [9]. Focal plane array (FPA) detector, FPA-FTIR microspectroscopy gains a significant advantage over micro-Raman spectroscopy by enabling the rapid chemical mapping of large sample areas. This high-speed acquisition is a crucial benefit, as it also inherently avoids the strong fluorescence interference often plaguing biological samples, a common limitation of micro-Raman spectroscopy. Although micro-Raman spectroscopy is better suited for analyzing wet samples without air drying, FPA-FTIR microspectroscopy is fast and lack of fluorescence issues make it an excellent tool for large scale biological and clinical studies [10,11].

FTIR microspectroscopy analyzes vibrational energy absorption in molecular bonds, providing a composite biochemical fingerprint of cells. The technique enables rapid and label-free quantification of major cellular components—including proteins, lipids, carbohydrates, and nucleic acids—at single-cell resolution. Importantly, changes in cell state are accompanied by alterations in molecular composition and structure, which can be detected in specific IR absorption bands. For example, shifts in the amide I and II regions (1700–1500 cm^−1^) reflect differences in protein secondary structure and abundance; CH_2_ and CH_3_ stretching modes (3000–2800 cm^−1^) are indicative of membrane lipid composition and fluidity; and bands in the 1200–800 cm^−1^ region reflect changes in nucleic acid conformation and RNA/DNA content [9,12,13,14]. Previous bio-spectroscopic studies have successfully applied FTIR to distinguish undifferentiated from differentiated cells, to assess stem cell lineage commitment, and to monitor reprogramming efficiency [9]. However, no study has explored its application for distinguishing between naïve and primed states of pluripotency, particularly in non-human primates. Establishing such a methodology could provide a powerful tool for monitoring cell identity and quality in basic and translational stem cell research. A technique known as FPA infrared imaging with multichannel detectors has recently become available for analyzing cells and tissues with allowing to measure all data points from each detector element simultaneously [9].

In this study, rhesus macaque embryonic stem cells (rhESCs) cultured under two distinct conditions: the well-established FGF2/KOSR medium, which supports the primed state, and a novel medium we developed, ALGöX [5], designed to support a naïve-like pluripotent state. The combination of ALGöX aims to inhibit MEK/ERK and WNT signaling while maintaining key self-renewal cues, based on pathways shown to stabilize naïve pluripotency in rodent and human systems. Cells from both conditions were analyzed by immunocytochemistry, RNA sequencing and FPA-FTIR microspectroscopy. This study represents the first model to utilize FPA-FTIR microspectroscopy for the purpose of characterizing and distinguishing primed from naïve-like rhESCs. This technique was selected due to its inherent advantages, including its non-invasive, high accuracy, and cost-effectiveness.

## 2. Results

### 2.1. Characterization of rhESC-FGF2/KOSR and rhESC-ALGöX Cells by Immunocytochemistry and RNA Sequencing

rhESCs were originally derived in FGF2/KOSR [15]. Morphological changes were observed when rhESCs-FGF2/KOSR were transferred into ALGöX medium. rhESCs formed more compact colonies (Figure 1a).

Immunolabeling confirmed that both rhESC-FGF2/KOSR and rhESCs-ALGöX retained expression of core pluripotency markers OCT4, NANOG, and SOX2 (Figure 1b), indicating stable pluripotent identity across conditions. In rhESC-FGF2/KOSR, the intensity of these markers peaked at passage 60 and declined progressively at later passages (65 and 70). In contrast, rhESC-ALGöX maintained consistent expression levels across passages (Figure 1c).

Notably, the primed-state marker OTX2 was detected exclusively in rhESC-FGF2/KOSR, while rhESC-ALGöX cells exhibited robust expression of KLF17, ALPPL2, TFCP2L1, and TFAP2C, consistent with a naïve pluripotency signature (Figure 1b). Quantitative fluorescence analysis revealed statistically significant differences in marker expression over passages, with the highest levels of OTX2 in primed cells at passage 65 and peak expression of naïve markers in rhESC-ALGöX cells at passage 32 (Figure 1c). These results indicate that rhESCs cultured in ALGöX medium maintain a naïve-like molecular profile and support stable naïve pluripotency over multiple passages.

RNA-seq confirmed the transcriptional divergence between rhESC-FGF2/KOSR and rhESCs-ALGöX. Hierarchical clustering of Z-score-normalized expression profiles segregated all samples into two distinct clusters corresponding to their respective culture conditions (Figure 2). rhESCs-FGF2/KOSR showed elevated expression of primed-state genes including NODAL, OTX2, ETV4, BMP4, FST, and SOX3, whereas rhESC-ALGöX exhibited high expression of naïve-associated genes including KLF2, DPPA2, DPPA3, ZFP42, PRDM14, and TFCP2L1. These transcriptomic data validate the immunolabeling data and strongly suggest that ALGöX conditions reprogram primed rhESCs to a molecularly distinct, naïve-like pluripotent state.

### 2.2. Characterization of rhESC-FGF2/KOSR and rhESC-ALGöX Cells: FTIR Microspectroscopy Distinguishes Primed and Naïve-like rhESCs

#### 2.2.1. Biochemical Differences Between Cell States

We used an FPA-based FTIR imaging system to analyze rhESC-FGF2/KOSR and rhESC-ALGöX cells at the high pixel resolution (9.6 × 9.6 µm). Three biological replicates per condition were analyzed: passages 60, 65, and 70 for primed rhESC-FGF2/KOSR, and passages 22, 27, and 32 for naïve-like rhESC-ALGöX. The amide I (1700–1600 cm^−1^) and amide II (1600–1500 cm^−1^) bands provide insight into protein secondary structures (Figure 3a). The amide I band, arising primarily from C=O stretching, resolves into features attributed to α-helices (~1654 cm^−1^), β-sheets (~1635 cm^−1^), and β-turns (~1685 cm^−1^) [9]. The amide II band originates from N–H bending and C–N stretching and typically peaks around 1546 cm^−1^. To resolve overlapping peaks, we applied second derivative spectroscopy, which enhanced the identification of individual vibrational bands [16]. The second derivative spectra of primed and naïve-like cells are shown in Figure 3b.

Naïve-like rhESCs-ALGöX displayed significantly higher integrated absorbance in the amide I, amide II, and nucleic acid regions compared to primed rhESC-FGF2/KOSR cells (*p* < 0.05, ANOVA; Figure 3c). Notably, naïve-like cells exhibited prominent bands at 1654 and 1546 cm^−1^, corresponding to Amide I and II, and strong nucleic acid peaks at 1239 and 1085 cm^−1^. In contrast, primed cells showed stronger lipid-associated absorbance, particularly in the CH_2_/CH_3_ stretching region (2921–2852 cm^−1^) and the ester carbonyl stretch at 1741 cm^−1^ (Figure 3b). These spectral differences suggest higher protein synthesis and nucleic acid content in naïve-like cells, while primed cells are enriched in membrane lipid signatures [17,18,19]. No statistically significant spectral variation was observed between passages within either state (Figure 3c).

#### 2.2.2. Principal Component Analysis of rhESCs Based on FTIR Spectra

PCA was performed on the second derivative spectra from rhESC-FGF2/KOSR and rhESC-ALGöX cells in order to visualize of clustering of similar spectra within datasets in scatter plots; and identification of variables (spectral bands representing various molecular groups within the samples) in loading plots. Unsupervised PCA of the full spectral dataset revealed clear segregation between cell states. PC−1 (64% variance) and PC−2 (4% variance) of the total variance effectively separated naïve-like rhESC-ALGöX and primed rhESC-FGF2/KOSR cells (Figure 4a). Loadings on PC−1 indicated strong contributions from lipid-associated bands (3000–2800 cm^−1^), Lipid ester carbonyl (1750–1700 cm^−1^), protein Amide I and Amide II-related bands (1700–1500 cm^−1^), Phosphodiester bond from nucleic acid (1240 and 1080 cm^−1^) (Figure 4b). Primed rhESC-FGF2/KOSR cells showed positive PC−1 scores associated with negative loading from CH-stretching (2852 and 2919 cm^−1^) and ester carbonyls (1739 and 1704 cm^−1^), while negative score plot from naïve-like rhESC-ALGöX cells were associated with positive loading from strong amide I absorption (~1652 cm^−1^), amide II absorption (~1544 cm^−1^) and phosphodiester bond from nucleic acid at 1232 and 1083 cm^−1^ (Figure 4b and Table 1).

#### 2.2.3. Partial Least Squares Discriminant Analysis (PLS-DA) of rhESCs Based on FTIR Spectra

Primed or naïve-like spectra from each group were randomly separated into calibration and validation sets, comprising approximately two-thirds and one-third of spectra, respectively. A total of 1581 spectra from primed cells and 3317 spectra from naïve-like cells were used for the analysis. The calibration data matrix employed for PLS-DA consisted of the spectral dataset (multivariate X) and two Y variables with integer values of 0 or 1 coding for the each of the two modeled spectral classes.

Classification of the dataset was then carried out by predicting a Y value for each spectrum in the independent validation using PLS models that had been generated from the calibration sets. Correct classification of each class was arbitrarily assigned to samples with predicted Y > 0.5 for respective spectra. The slope of the regression line is 0.955, which is very close to the ideal value of 1.0. The offset is 0.014, which is close to the ideal value of 0. This suggests a strong and consistent linear relationship between the predicted and measured values. The resulting model demonstrated strong predictive performance, with a correlation coefficient of *R* = 0.95. This indicates a high degree of agreement between the predicted and measured values, which is crucial for practical applications. The low RMSE of 0.098 ensures that the magnitude of the average prediction error is minimal and well within acceptable limits, thus confirming the model is suitability and precision for use (Figure 5a,b). The classification accuracy in identifying primed and naïve-like samples reached 100% specificity and 100% sensitivity.

## 3. Discussion

Our study demonstrates that FTIR microspectroscopy is a robust, label-free, and non-invasive approach for distinguishing primed and naïve-like states in rhesus macaque ESCs. Focal plane array (FPA) detectors use multiple elements that allow for the simultaneous measurement of all data points from each detector element in the spectral interval recorded, with each detector pixel recording independently. Each detector pixel functions as an aperture and records the entire spectrum. Using FPA detector, we identified reproducible spectral signatures that correspond to characteristic differences in protein, lipid, and nucleic acid composition between the two pluripotent states [13,22]. To ensure the highest quality spectral data, all samples were measured in transmission mode using a Barium Fluoride (BaF_2_) slide. This method was chosen specifically to eliminate potential dispersion artifacts that can occur in reflection mode measurements. Furthermore, we can confirm that Mie scattering did not affect our results, as no band distortions were observed in the Amide I and Amide II regions. To further prevent such distortions, the intensity of the Amide I band was kept below 1.0 absorbance unit [23,24]. We found that rhESCs cultured in ALGöX exhibit distinct spectral features compared to those in FGF2/KOSR.

Naïve-like rhESCs cultured in ALGöX medium exhibited stronger absorbance in the amide I (~1654 cm^−1^) and amide II (~1546 cm^−1^) regions. Prominent peaks at 1240 and 1080 cm^−1^, attributed to phosphodiester groups in nucleic acids, suggest increased transcriptional activity and molecular complexity. These biochemical features align with the molecular hallmarks of naïve pluripotency, including greater developmental plasticity and retention of an earlier embryonic identity [25]. In contrast, primed rhESCs maintained in FGF2/KOSR medium displayed stronger lipid-associated absorbance, including CH_2_/CH_3_ stretching vibrations (2921–2852 cm^−1^) and ester carbonyl peaks (1741 cm^−1^). Similar lipid enrichment has been reported in primed human ESCs analyzed by FPA-FTIR microspectroscopy, where high CH_2_/CH_3_ intensity was linked to membrane remodeling and metabolic adaptations [17]. Metabolically, naïve PSCs rely predominantly on oxidative phosphorylation, whereas primed PSCs shift toward glycolysis and increased lipid utilization, which may underlie the lipid signatures observed here [18,19].

Multivariate analyses reinforced these findings: PCA revealed clear separation of primed and naïve-like spectra, while PLS-DA achieved 100% specificity and 100% sensitivity in classifying cell states. These FTIR-based distinctions were fully consistent with immunocytochemistry—showing mutually exclusive expression of OTX2 in primed cells and KLF17, TFCP2L1, ALPPL2, and TFAP2C in naïve-like cells—and with transcriptomic data that confirmed state-specific gene expression patterns [5,25,26].

This work represents, to our knowledge, the first FTIR-based biochemical profiling of primed and naïve pluripotent states in a non-human primate model. It establishes FTIR microspectroscopy as a new approach for pluripotent stem cell phenotyping at the single-cell level [13,27]. A key question is whether this approach can reliably distinguish the naïve pluripotent state induced under diverse culture conditions and in multiple species. The discovery of a conserved and distinctive molecular signature would greatly enhance our ability to identify and validate the naïve state across experimental systems. With developing and validating PLS-DA models based on much higher sample numbers this technique might be further tested and ultimately applied as a practical tool for optimization for identifying stem cells. In our opinion, this study represents the first steps toward achieving this aim. We employed the QUASAR 1.11.1 software [28] for cell classification. The software was selected for its ability to analyze a large number of spectra—specifically, more than 3000 spectra simultaneously—a significant advantage over the Unscrambler X 10.3 software (CAMO, Oslo, Norway), which has limitations in this regard. The results demonstrated that the QUASAR software achieved 99% specificity and 99% sensitivity (Appendix A), matching the performance of Unscrambler X. Furthermore, the QUASAR software offers platform flexibility and high processing speed. For future work, we plan to compile a comprehensive database of spectra from various types of stem cells and integrate machine learning for cell classification. This approach is intended to substantially reduce the high costs, complex procedures, and extensive time currently associated with traditional analysis methods.

## 4. Materials and Methods

### 4.1. Preparation of Feeder Cells

Mouse embryonic fibroblasts (MEFs) were isolated from 13.5-day-old OF1 mouse embryos (Charles River, Lyon, France) following the protocol previously described [29]. MEFs were cultured in MEF medium consisting of Dulbecco’s modified eagle medium (DMEM, Gibco, Paisley, UK) supplemented with 10% fetal bovine serum (FBS, Gibco), 1% Non-essential amino acids (NEAA, Gibco), 1% penicillin-streptomycin-glutamine (PSG, Gibco). Cells were incubated at 37 °C in a humidified atmosphere of 5% CO_2_ in air. Cells were passaged with trypsin-EDTA (Gibco) and frozen at passage 2 (P2) in medium containing 10% dimethyl sulfoxide (DMSO, Sigma-Aldrich, St. Louis, MO, USA). Frozen-thawed MEFs were cultured to 90% confluence and mitotically inactivated by incubation in fresh MEF medium supplemented with 5 µg/mL mitomycin C (Sigma-Aldrich) at 37 °C for 3 h. Cells were then washed five times with Ca^2+^/Mg^2+^ free PBS (Gibco), trypsinzed, incubated for 5 min at 37 °C, centrifuged at 400× *g* for 5 min, and resuspended in MEF medium. Mitomycin-treated MEFs (2.5 × 10^5^ cells) were plated in 35 mm culture dishes before seeding rhESCs.

### 4.2. Preparation of Conditioned Medium (CM)

MEFs (4 × 10^6^) were plated in 100 mm dishes (SPL Life Sciences, Gyeonggi, Republic of Korea) coated with 0.1% gelatin. One day after plating, MEF medium was replaced with 25 mL N2B27 basal medium containing 48.7% DMEM/F12 (Gibco), 48.7% Neurobasal medium (Gibco), 1% B27 supplement (Gibco), 0.5% N2 supplement (N2, homemade), 0.02% β-mercaptoethanol (Gibco), 1% PSG, and 20 ng/mL FGF-basic human (bFGF, Sigma-Aldrich). After 24 h, the N2B27 conditioned medium (N2B27-CM) was collected and replaced with an equal volume of fresh complete N2B27 medium. Conditioned medium was collected daily for three consecutive days, pooled, filtered, and stored at −20 °C until use.

### 4.3. Culture and Expansion of rhESCs-FGF2/KOSR

rhESCs-FGF2/KOSR cells were cultured on feeders in 35 mm dishes with 2 mL ESC medium containing 80% KO-DMEM (Gibco), 20% knockout serum replacement (KOSR, Gibco), 1% NEAA (Gibco), 1% GlutaMAX (Gibco), 0.1 mM β-mercaptoethanol (Sigma-Aldrich), and 5 µg/mL bFGF (Sigma-Aldrich). Frozen-thawed rhESCs were cultured in medium supplemented with 10 µM ROCK inhibitor (Y-27632) (TOCRIS Bioscience, Bristol, UK) for the first 24 h. Cultures were maintained at 37 °C, 5% CO_2_ and 5% O_2_, with daily medium changes. Cells were passaged every 3–4 days by mechanical dissociation.

### 4.4. Culture and Expansion of rhESCs-ALGöX

rhESCs-ALGöX cells were cultured in 35 mm dishes pre-coated with 5 µg/mL laminin (LN521, STEMCELL Technologies, Lund, Sweden) for 1 h. Cells were maintained in 2 mL ALGöX medium [5] consisting of N2B27-CM supplemented with 10 ng/mL Activin A (Peprotech, Cranbury, NJ, USA), 1000 U/mL LIF (homemade), 1.25 μM Gö6983 (TOCRIS Bioscience), and 2.5 μM XAV939 (Sigma-Aldrich). Cultures were maintained at 37 °C, 5% CO_2_ and 5% O_2_, with daily medium changes until 80% confluence. Cells were passaged every 3–4 days using 1X TrypLE (Gibco) for single-cell dissociation, followed by addition of 10 µM Y-27632 for the first 24 h to promote cell survival.

### 4.5. Immunocytochemistry and Imaging

rhESCs-FGF2/KOSR cells and rhESCs-ALGöX cells were fixed with 4% paraformaldehyde (PFA, Sigma-Aldrich) in PBS for 20 min at room temperature, washed twice with PBS (5 min each), and permeabilized with PBS containing 0.5% Triton X-100. Non-specific binding sites were blocked in PBS with 10% donkey serum for 1 h at room temperature. Cells were incubated overnight at 4 °C with primary antibodies (Appendix A). After three washes, cells were incubated with secondary antibodies (Appendix A) for 1 h at room temperature. Nuclei were stained with DAPI (0.5 µg/mL), and cells were mounted with M1289 mounting medium (Sigma-Aldrich). Confocal imaging was performed using a DM 6000 CS SP5 microscope (Leica, Wetzlar, Germany) with a 45×/1.25 oil immersion objective (PL APO HCX, Leica) [8,25].

### 4.6. RNA Sequencing

Total RNA was extracted from 4–5 × 10^6^ cells using the RNeasy mini-kit (Qiagen, Hilden, Germany). Libraries were prepared from 200 ng of RNA using the NextFlex Rapid Directional mRNA-Seq kit (Bio-Scientific, Boston, MA, USA) and sequenced on a NextSeq500 platform (Illumina, San Diego, CA, USA) as single end 75 bp reads. Demultiplexing was performed with bcl2fastq (Illumina), and adapters were trimmed using Cutadapt. Sequencing depth was ~30 million reads per sample. Reads were aligned to the reference genome using HISAT2, and gene counts were generated with HTSeq. RNA-seq data from this study are available under GEO accession number GSE146178 [7,26].

### 4.7. FPA-FTIR Microspectroscopy

Cell pellets were washed three times with 0.9% NaCl, resuspended in 50 µL saline, and 5 µL aliquots were deposited onto IR-transparent 2 mm thick barium fluoride (BaF_2_) windows. Samples were air-dried and stored in a desiccator until analysis [30]. Spectral data was acquired from an FPA-FTIR microscope (Hyperion 3000) with a FPA detector, connected to Tensor 27 FTIR spectrometer (Bruker Optics, Ettlingen, Germany) at the Synchrotron Light Research Institute (Public Organization), Nakhon Ratchasima, Thailand. The acquisition parameters were a 36× objective lens in transmission mode, 64 scans and all data was measured with 8 × 8 binning at a spectral resolution of 6 cm^−1^ [31]. The full area of 64 × 64 pixels equal 4096 pixels. The field of view area using a 36× objective lens was 70.4 × 70.4 μm^2^. The image area composed of 3 × 1 frames (211.2 × 70.4 μm^2^). An 8×8 binning resulted in a pixel resolution of 8.8 μm^2^ and obtain 2000 to 3000 spectra from each sample. The biochemical composition distribution was performed by OPUS 7.5 software (Bruker Optics, Ettlingen, Germany).

### 4.8. Multivariate Data Analysis of FTIR Spectra

Spectral quality control was performed by visual inspection. Spectra with weak absorbance (amide I band maximum absorbance < 0.2), acquired from regions of the sample where there were no cells, or spectra with very high absorbance (amide I band maximum absorbance > 0.8), acquired from regions where cells may have been clumped or overlaid were rejected from the analysis. Spectra from sample groups based on differentiation of cell states and passage cells were analyzed using PCA. Preprocessing of the data was conducted by first performing a second derivative by the Savitzky—Golay method (13 smoothing points) and then normalized using EMSC with the spectral regions from 3000 to 2800 cm^−1^ and from 1800 to 800 cm^−1^ using The Unscrambler X 10.3 software. Score plots were used to visualize any clustering of the data, and the loading plot was used to determine which spectral regions contributed most to the variance in the dataset, accounting for any clustering of spectra seen in scores plots. Integrated peak areas were analyzed by OPUS 7.5 software. Spectra were analyzed using PLS-DA by The Unscrambler X 10.3 software. PLS-DA employed PCA models derived from calibration sets and was used to test the ability to discriminate different cell states using the independent validation set spectra. Calibration and validation of spectral data were employed using datasets that were randomly selected, comprising two-thirds and one-third of the spectra, respectively. The dataset could be utilized to calculate the percentages of specificity and sensitivity. FTIR microspectroscopy combined with multivariate data analysis, in particular PCA, was applied to explain biochemical changes occurring during cellular differentiation.

### 4.9. Statistical Analysis

All statistical analyses were performed using GraphPad Prism version 5 (GraphPad Software, San Diego, CA, USA). Data are presented as Standard Error of the Mean (SEM). Differences among groups were analyzed using one-way analysis of variance (ANOVA), followed by Tukey–Kramer’s Honest Significant Difference (HSD) post hoc test for pairwise comparisons. A *p*-value of less than 0.05 was considered statistically significant. Different letters (a, b, c, d, e) above the bars indicate a statistically significant difference (*p* < 0.05) between the groups. Identical letters indicate no significant difference. Graphs were generated using Sigma Plot version 15 (Grafiti LLC, Palo Alto, CA, USA).

## 5. Conclusions

This work presents the first application of FTIR microspectroscopy to characterize and distinguish primed from naïve-like rhESCs. The FTIR-based results were cross-referenced using two established approaches—immunocytochemistry and RNA sequencing—which confirmed that naïve-like cells contain higher protein levels, whereas primed cells are enriched in lipids. Each method provided complementary insights: immunocytochemistry localized protein expression, RNA sequencing offered gene-level resolution, and FTIR delivered a rapid, cost-effective overview of cellular biochemical composition. Recognizing the potential of this approach, we anticipate that integrating large-scale datasets with machine learning will be essential for its full development. Such a synergy could enable near-instantaneous data analysis and result generation, potentially transforming the field.

Beyond rhESCs, this strategy holds broad promise. It could be applied to monitor differentiation in diverse cell types and serve as a rapid diagnostic platform, bridging the gap between genomic information and cellular phenotypes. In clinical contexts, FTIR microspectroscopy could serve as a real-time, non-invasive screening tool for quality control in regenerative medicine, ensuring the purity and developmental state of cell products and thereby enhancing their safety and efficacy.

## Figures and Tables

**Figure 1 ijms-26-09514-f001:**
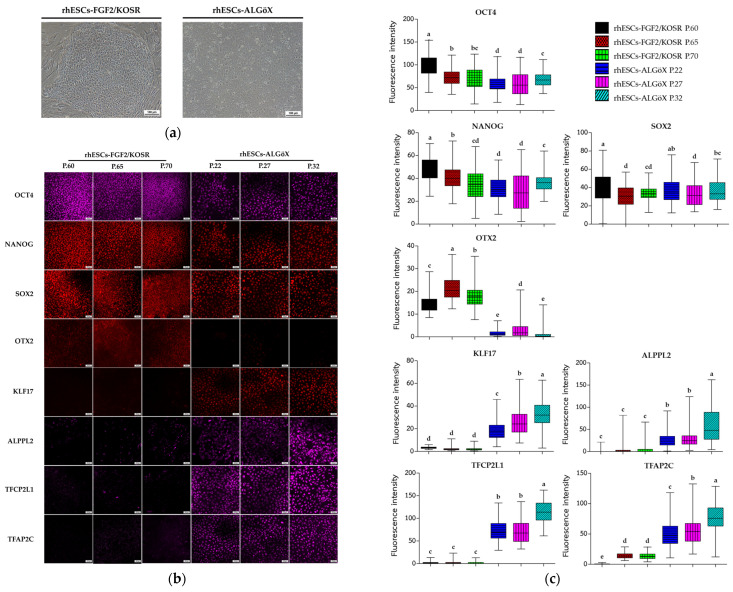
Characterization of rhESC-FGF2/KOSR and rhESC-ALGöX cells. (**a**) Phase-contrast images of rhESCs cultured in FGF2/KOSR and ALGöX. Scale bars: 100 µm. (**b**) Immunolabeling of rhESCs cultured in FGF2/KOSR and ALGöX. Core pluripotency markers: OCT4, NANOG, and SOX2. State-specific markers: OTX2 (primed) and KLF17, ALPPL2, TFCP2L1, and TFAP2C (naïve-like). Scale bars: 100 µm. (**c**) Quantification of fluorescence intensity in rhESCs cultured in FGF2/KOSR or ALGöX following immunolabeling. Different letters (a, b, c, d, e) above the bars indicate a statistically significant difference (*p* < 0.05) between the groups.

**Figure 2 ijms-26-09514-f002:**
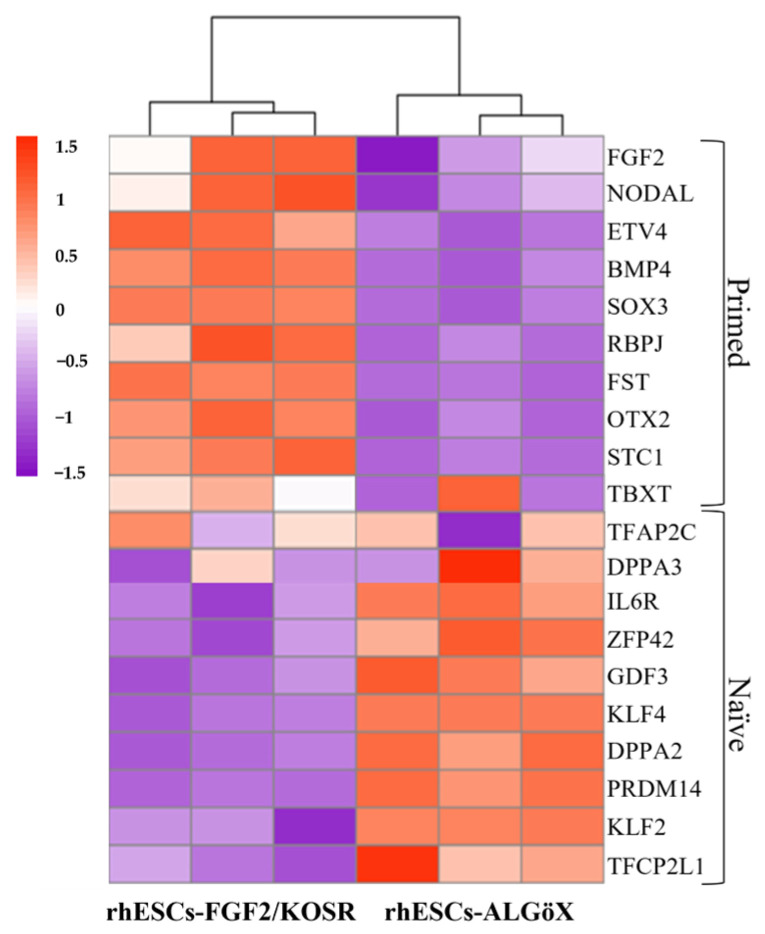
Heatmap of gene expression profiles in rhESCs cultured in FGF2/KOSR or ALGöX based on RNA sequencing data. Genes associated with primed and naïve pluripotency states are shown. Scale bar indicates log10 of mRNA quantification ranging from −1.5 (low expression) to 1.5 (high expression).

**Figure 3 ijms-26-09514-f003:**
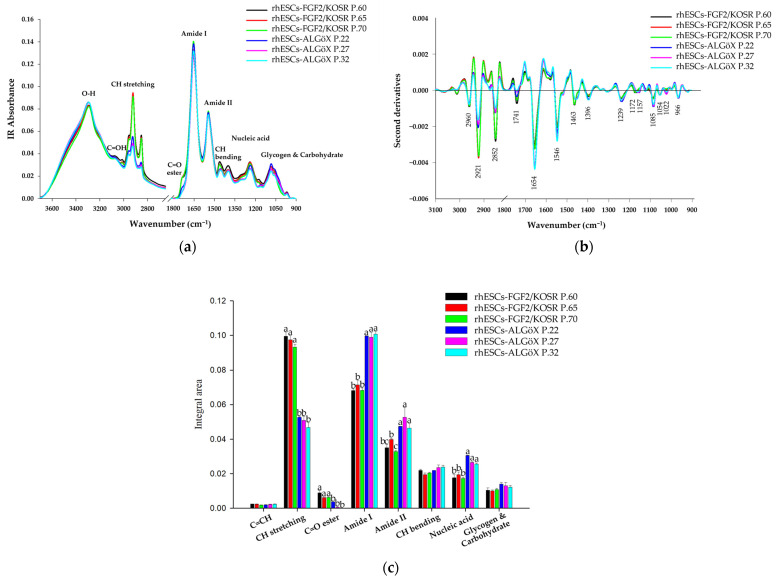
FPA-FTIR microspectroscopy of rhESCs cultured in FGF2/KOSR or ALGöX. (**a**) Smoothed (13-point) and normalized absorbance spectra (4000–800 cm^−1^). (**b**) Second-derivative spectra normalized by extended multiplicative signal correction (EMSC) for the 3000–2800 cm^−1^ and 1800–800 cm^−1^ regions. (**c**) Histograms of relative integrated areas of macromolecular components from normalized second-derivative spectra (OPUS 7.5). Different letters (a, b, c) above the bars indicate a statistically significant difference (*p* < 0.05) between the groups.

**Figure 4 ijms-26-09514-f004:**
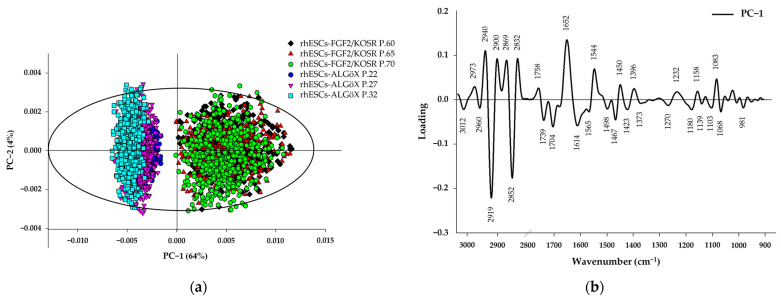
Principal Component Analysis of rhESCs based on FTIR spectra. (**a**) PCA of the full spectral range (800–4000 cm^−1^). (**b**) PC−1 loading plots from independent spectra.

**Figure 5 ijms-26-09514-f005:**
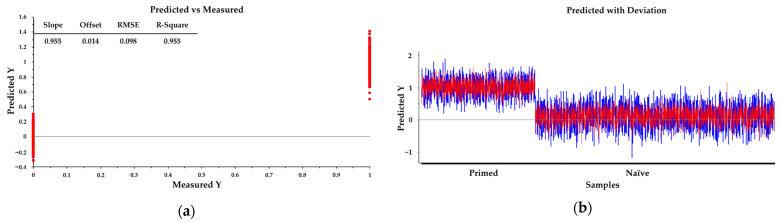
PLS-DA modeling of rhESCs based on FTIR spectra. (**a**) Calibration (training) set showing measured versus predicted Y values, with primed state = +1 and naïve-like state = 0. (**b**) Validation set predictions using the PLS-DA model.

**Table 1 ijms-26-09514-t001:** Band maxima distinguishing primed and naïve-like pluripotent states in rhESCs as identified by FPA-FTIR microspectroscopy.

Band Maxima Second Derivative Spectra (cm^−1^)		PC−1 Loading (cm^−1^)		Band Assignments
rhESCs-FGF2/KOSR	rhESCs-ALGöX	Negative Loading	Positive Loading	
P60, P65, P70	P22, P27, P32			
2960	2960		2940	CH_3_ asymmetric stretch due to methyl terminal of membrane phospholipids: mainly lipids [9,17,20,21]
2921	2921	2919	2900	CH_2_ asymmetric stretch of the methylene group of membrane phospholipids: mainly lipids, with some contribution from proteins, carbohydrates, nucleic acids [9,17,18]
2852	2852	2852	2869, 2832	CH_2_ symmetric stretching: mainly lipids, with some contribution from proteins, carbohydrates, nucleic acids [9,17,20,21]
1741	1741	1739, 1704		C=O stretching vibrations of lipids (triglycerides and cholesterol esters) [9,17,20,21]
1654	1654	1614	1652	Amide I: C=O (80%) and C—N (10%) stretching, N—H (10%) bending vibrations: proteins α-helix [9,17,21]
1546	1546		1544	Amide II: N—H (60%) bending and C—N (40%) stretching vibrations: proteins α-helix [17,21]
1463	1463	1467	1450	CH_2_ bending vibrations: lipids and proteins [17] Cholesterol methyl band [21]
1396	1396		1396	COO− stretching vibrations of amino acid side chains [9,20,21]
1239	1239		1232	PO_2_-asymmetric stretching vibrations: RNA, DNA, and phospholipids [9,17,20,21]
1172	1157, 1022	1180	1158	C–O–C vibrations from glycogen and other carbohydrates [9,17,20,21]
1085	1085		1083	PO_2_-symmetric stretching vibrations: RNA, DNA [9,17,20,21]
1054	1054	1068		C—O vibrations from glycogen and other carbohydrates [9,17]
966	966	981		C—O deoxyribose, C—C DNA [21]

## Data Availability

The data presented in this study are available on request from the corresponding author.

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
