# Peer review of "Characterization of Rhesus Macaque Embryonic Stem Cells in Primed and Naïve-like Cell States of Pluripotency Using Fourier Transform Infrared (FTIR) Microspectroscopy"

_ijms, 2025, doi:10.3390/ijms26199514_

Round 1
Reviewer 1 Report
Comments and Suggestions for Authors
The authors evaluated the application of Fourier-transform infrared (FTIR) microspectroscopy for the non-invasive biochemical profiling of rhesus macaque embryonic stem cells (rhESCs) cultured in either conventional FGF2/KOSR medium or a novel formulation, ALGöX, as well as two other conventional biological techniques.
By means of these two other techniques, the authors have already established that rhESCs maintained in FGF2/KOSR were in the pluripotent state, whereas those cultured in ALGöX adopted a naïve-like state. FTIR spectra showed consistently different protein, lipid, and nucleic acid signatures, as well as in the amide I/II and nucleic acid absorbance, between the cells cultured in FGF2/KOSR medium or the ALGöX formulation.
Principal component analysis (PCA) of the FTIR spectra, followed by partial least squares-discriminant analysis (PLS-DA), allowed classification of the groups with 100% specificity and 100% sensitivity. The result is indeed conclusive, and leaves no room for any controversy about the convenience of the applicability of FTIR to distinguish pluripotent stem cells (PSCs) from those that may be just in a naïve-like state. The separation of the two groups of cells illustrated in Fig. 4a is also conclusive and an example of the convenience of the applicability of FTIR for future studies.
Hence, I have no further commentary. I think this is a piece of fine scientific work and has all the merits to be published in its present form.
Reviewer 2 Report
Comments and Suggestions for Authors
The article entitled “Characterization of Rhesus Macaque Embryonic Stem Cells in Primed and Naïve-like cell States of Pluripotency Using Fourier Transform Infrared (FTIR) Microspectroscopy” refers to the application of Focal Plane Array (FPA) FT-IR spectroscopy for identifying and distinguishing rhesus macaque embryonic stem cells (rhESC) rhESC-FGF2/KOSR from rhESC-ALGöX. The results from FPA/FT-IR spectroscopy are also supported by results from immunocytochemistry and RNA sequencing. Although the manuscript is well-structured and the importance of the topic is high, a couple of issues should be addressed:
- Lines 77-79: Please present references for FPA-IR imaging.
- Lines: 83-88: They should be transferred to Material and Methods section.
- Lines: 88-89: They should be transferred to Results or Discussion section.
- The aim of the study and its novelty is missing from the Introduction of the study. Please present clearly the aim and the objectives of the study. Also, please explain thoroughly the reason why they have selected FPA-IR imaging except for other techniques. For example, micro-Raman spectroscopy could also combine hyperspectral imaging, while reducing the interference from water, which is an important aspect for biological samples.
- Figure 1c: Please explain in the caption the labels “a-e”.
- Figure 2: Please present in the figure what the scale bar of the heatmap shows.
- Lines 143-144: Please provide reference for the selection of the second derivative for resolving overlapping peaks.
- Figure 3c: The p-values and stars of significant difference should be presented on the figure.
- Figure 4a: Please add confidence interval ellipses on the data and mention the confidence interval applied.
- Table 1: Two PC1 loading wavenumbers are missing and 1158,1180 cm-1 is not in the right order (descending wavenumbers).
- Lines 199-200: Please rephrase the sentence because of syntax error.
- Figure 5a: Please take care of the significant digits in the legend of the graph and use a greater font size. Also, please explain the titles of the x- and y-axis (Primed2.3 and Factor-7) in the caption of the figure.
- Lines 243, 244, 246, 247, 248: Please replace “program” with “software”.
- Line: 245-246: Please add a reference for Unscrambler X software.
- Subsection 4.7: Please add the NA of the objective lens, as well as the spatial resolution and the total number of spectra acquired from each sample.
- Line 318: Air-drying prior to analysis could lead to decomposition or degradation of biological samples, cell death, biochemical changes in proteins and nucleic acids. Please provide the experiments proving that air-drying has no significant effect on the sample stability during analysis.
- Subsection 4.8: Please have a careful grammar check of the whole subsection.
- Line 350: There are no mean and standard deviations in the manuscript. Please amend accordingly.
- Line 359: Validation includes determination of accuracy, precision, specificity, sensitivity and robustness of the method. Please modify the word “validated” so as not to confuse the reader.
- Conclusions section: The conclusion section does not emphasize the novelty of the study and does not lead to suggestions for possible applications of the results.
- Abbreviations section: cm-1 is not an abbreviation, it is a unit. α-helix, β-sheet, C=O stretching, C-N stretching, CH stretching should be removed from abbreviations, as they are commonly used in the biochemistry field.
Please take into consideration the syntax and the grammar of the aforementioned sections.
Reviewer 3 Report
Comments and Suggestions for Authors
As a proof-of-concept study, the work " Characterization of Rhesus Macaque Embryonic Stem Cells in Primed and Naïve-like cell States of Pluripotency Using Fourier Transform Infrared (FTIR) Microspectroscopy" demonstrates that FTIR microspectroscopy can distinguish cells maintained under conditions that induce naïve versus primed pluripotent states. I find the approach quite innovative, and the results are promising However, a few considerations could further strengthen the study:
1 The current findings are based on a single cell line, which limits the generalizability of the conclusions. To better establish the robustness of FTIR-based classification, it would be valuable to validate the results across multiple independent cell lines.
2 The PLS-DA model requires rigorous external validation on independent samples, for example, naïve-state cells derived using a different protocol or from a different cell line to demonstrate that the model generalizes well and is not overfitting to idiosyncratic features of the current dataset.
3 Full confidence that the observed spectral differences reflect genuine biochemical changes, rather than artifacts caused by Mie scattering, necessitates the use of specialized algorithms for resonant Mie scattering correction. A more detailed discussion of this technical challenge and its potential impact on the data interpretation would strengthen the manuscript.
Round 2
Reviewer 2 Report
Comments and Suggestions for Authors
The manuscript entitled “Characterization of Rhesus Macaque Embryonic Stem Cells in Primed and Naïve-like cell States of Pluripotency Using Fourier Transform Infrared (FTIR) Microspectroscopy” was improved by the authors after answering most of the initial comments. However, some points remain still to be addressed, especially about the selection of the FTIR technique and the required sample preparation (air-drying) prior to the analysis. The comments follow:
- Regarding the initial comment 4, the authors have already improved the introduction section; however, the advantages of the applied method (FPA-FTIR) to micro-Raman spectroscopy are missing. Micro-Raman spectroscopy could be used to analyze biological samples, such as cells, which have high content of water without the need of air-drying, which could lead to decomposition of the samples. Please provide the appropriate explanations with respective references.
- Regarding the initial comment 12, the statistical data of the model should be retained (at least R2, RMSEC, RMSECV, slope), but the authors should take care of the significant digits for each one.
- Regarding the initial comment 15, the numerical aperture (NA) of the objective lens should be added in the respective section of Material and Methods section.
- Regarding the initial comment 16, the experiments conducted to prove stability of the sample after air-drying should be presented in the supplementary material. If it is a standard protocol described elsewhere, please provide the respective reference.
Reviewer 3 Report
Comments and Suggestions for Authors
While the authors have provided a detailed description of the issue of resonant Mie scattering correction and referenced methodological literature, it remains unclear from the manuscript text how these methods were specifically applied to their own dataset. Specifically, were the spectra explicitly checked for the presence of Mie scattering artifacts and subjected to specialized processing, such as using the RMieS-EMSC algorithm, to ensure that the observed biochemical differences are not a consequence of these technical artifacts?
Round 3
Reviewer 2 Report
Comments and Suggestions for Authors
The authors have addressed the concerns about the introduction and the design of the study. However, a minor revision is required for the NA (numerical aperture) of the objective lens as referred in my initial comment 15. The magnification of the lens was added (36x) by the authors but not the numerical aperture, which gives crucial information about the resolution of the microscope.
